

# Dirac spin liquid as an "unnecessary" quantum critical point on square lattice antiferromagnets

**Yunchao Zhang[1], Xue-Yang Song[1,2] and Todadri Senthil[1]**

**1** Department of Physics, Massachusetts Institute of Technology,
Cambridge MA 02139-4307, USA
**2** Department of Physics, Hong Kong University of Science and Technology,
Clear Water Bay, Hong Kong, China

## Abstract

Quantum spin liquids are exotic phases of quantum matter especially pertinent to many modern condensed matter systems. Dirac spin liquids (DSLs) are a class of gapless quantum spin liquids that do not have a quasi-particle description and are potentially realized in a wide variety of spin 1/2 magnetic systems on $2d$ lattices. In particular, the DSL in square lattice spin-1/2 magnets is described at low energies by $(2+1)d$ quantum electrodynamics with $N_f = 4$ flavors of massless Dirac fermions minimally coupled to an emergent $U(1)$ gauge field. The existence of a relevant, symmetry-allowed monopole perturbation renders the DSL on the square lattice intrinsically unstable. We argue that the DSL describes a stable continuous phase transition within the familiar Neel phase (or within the Valence Bond Solid (VBS) phase). In other words, the DSL is an "unnecessary" quantum critical point within a single phase of matter. Our result offers a novel view of the square lattice DSL in that the critical spin liquid can exist within either the Neel or VBS state itself, and does not require leaving these conventional states.

# 1 Introduction

Quantum spin liquids are a class of ground states of frustrated quantum magnets that are long range entangled, and are often described by theories with exotic features such as emergent gauge fields and associated 'fractionalized' degrees of freedom [1–3]. There are a wide variety of QSLs that are known to be possible theoretically with distinct universal properties. An especially important example is the $U(1)$ Dirac spin liquid (DSL)[1] in two space dimensions. The low energy physics of the DSL is described by massless QED$_3$, a theory of massless emergent Dirac fermions coupled to an emergent $U(1)$ gauge field [4–7]. The DSL does not admit a quasi-particle description and is instead described by an interacting conformal field theory (CFT) in the infrared. DSLs have been studied as candidate stable quantum spin liquids on a variety of lattices, see, eg, Refs. [7–17].

Σ In this paper we focus on the DSL state in square lattice spin-1/2 quantum magnets which has been studied extensively previously [7, 9, 17–20]. It has special interest [19] as a parent Mott insulator that naturally evolves into a nodal $d$-wave superconductor upon doping. It has also been studied [9] as a parent of many competing orders such as the conventional Neel and Valence Bond Solid (VBS) phases.

Here we present a new interpretation of the DSL fixed point in spin-1/2 square lattice antiferromagnets as a quantum critical point (QCP) that lies within the Neel phase, *i.e* as a Neel-Neel QCP. Thus it is an example of an 'unnecessary' QCP introduced in Ref. [21], which describes transitions within a single phase of matter, analogous to a liquid-gas transition, except that it is continuous. The QCP is unnecessary or avoidable in the sense that there exists a smooth adiabatic path in the phase diagram connecting the two phases on either side without crossing the QCP.

This is in contrast to a conventional QCP, which also has a single symmetry allowed relevant perturbation but tunes the phase transition between two *different* phases.

The first examples of unnecessary quantum critical points were found [21] in $(3 + 1)d$ in the phase diagram of certain non-abelian gauge theories coupled to enough massless matter fields so as to render them infra-red free and in some free fermion QCPs in both $2+1$ and $3+1$ dimensions. More examples, including some simple ones, of unnecessary QCPs were also found [21, 23–26] in other free fermion QCPs in $2 + 1$-D and in a variety of $(1 + 1)d$ systems. The identification of the square lattice DSL as an unnecessary QCP adds a particularly familiar theory to this list.

---

[1]The DSL used to be called the Algebraic Spin Liquid (ASL) in the literature. The terminology DSL emphasizes the origins of this state through a parton mean field description of the spin model where there are spinons with Dirac dispersion. The terminology ASL emphasizes the algebraic correlations of local operators which is a property of the low energy physics irrespective of the particular parton description. Indeed we should regard the low energy theory abstractly as described by an interacting conformal field theory with a particular global symmetry that has a particular 't Hooft anomaly.

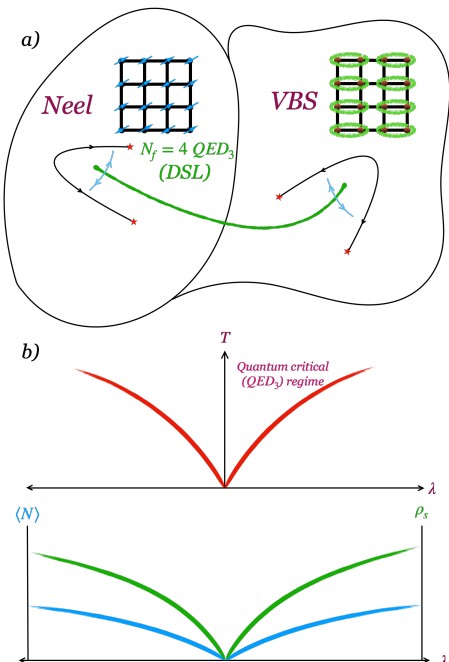

Figure 1: a) We exhibit QED$_3$ as an unnecessary quantum critical point within the Neel and VBS phases. The *unnecessary* quantum critical point is depicted here as a codimension-1 manifold (green line) in parameter space. There are paths, such as the ones shown, that avoid the QCP; whether it is possible to completely engulf the critical line or not is not clear but the plausible case is that the unnecessary critical manifold (a line in our two dimensional plot) intersects the Neel-VBS phase boundary, as depicted here. b) We represent the quantum critical fan, with $\lambda$ labeling the monopole fugacity. The QCP is located at $\lambda = 0$. In the second graph, we show the Neel order parameter $\langle N \rangle$ and spin stiffness $\rho_s$ as a function of $\lambda$, where the unnecessary QCP at $\lambda = 0$ can be accessed within the Neel phase. Both $\langle N \rangle$ and $\rho_s$ vanish as a power law of the correlation length $\xi$, with $\langle N \rangle \sim \xi^{-\Delta_N}$ and $\rho_s \sim \xi^{-1}$ from scaling arguments. Large $N_f$ calculations [9, 22] estimate $\Delta_N = 2 - \frac{64}{3\pi^2 N_f}$.

Remarkably, for the square lattice DSL, this means we do not need to exit the Neel phase to access it. Instead, the DSL can be accessed within the Neel state itself as an unnecessary QCP, which could shed new light on theories that build on this spin liquid to describe superconductivity in doped square lattice Mott insulators. We remark that the same DSL fixed point also describes a VBS-VBS phase transition (*i.e*, an unnecessary QCP) within the VBS phase. We exhibit these results in Figure 1.

## 2 Background on the Dirac spin liquid

Let us first quickly review how QED$_3$ arises as a low energy effective theory of a lattice spin-1/2 magnet on the square lattice. We begin with a parton decomposition of the lattice spin operator in terms of fermionic spinon operators:

$$\mathbf{S}_i = \frac{1}{2} f_{i,\alpha}^\dagger \boldsymbol{\sigma}_{\alpha\beta} f_{i,\beta} \,, \tag{1}$$

here $f_{i,\alpha}^\dagger$ is the spinon creation operator on site $i$ with spin $\alpha$. As is well-known, the representation Eq. (1) of the physical spin operator is redundant which is captured by an $SU(2)_g$

gauge constraint[2] to recover the physical Hilbert space [19]. Inserting this representation into a microscopic spin Hamiltonian and treating the resulting interactions within mean field theory leads to an effective quadratic lattice Hamiltonian. This mean field state may break the $SU(2)_g$ gauge group to some subgroup. A low energy theory for a possible phase (or critical point) of the original spin system is obtained by coupling the fermions in the mean field state to a dynamical gauge field transforming in the unbroken subgroup of $SU(2)_g$. We will be interested in the staggered flux state described by a mean field Hamiltonian of the form

$$H_{MF} = -\sum_{ij} f_i^\dagger t_{ij} f_j, \tag{2}$$

where $t_{i,i+\hat{x}} = (-1)^y t$ for vertical links and $t_{i,i+\hat{y}} = e^{i(-1)^{x+y}\theta/2} t$ on the horizontal links, with the flux $\pi \pm \theta$ for alternating plaquettes. This mean field Hamiltonian can be recast as the BCS Hamiltonian for a a $d_{x^2-y^2}$-wave superconductor of the spinons, and thus is also known as the d-wave Resonating Valence Bond (dRVB) state. This Mott insulating state thus has a close connection to a $d$-wave superconductor, and hence naturally evolves into the latter upon doping; for more detail, see Ref. [19].

The band structure of the staggered flux/dRVB mean field Hamiltomnian leads to $N_f = 4$ gapless Dirac nodes for the spinons. Furthermore the mean field ansatz breaks $SU(2)_g$ to $U(1)$ so that the low energy theory must include a dynamical $U(1)$ gauge field. We thus arrive at the QED$_3$ theory with $N_f = 4$ massless Dirac fermions as a low energy effective field theory of this state. The corresponding Lagrangian is

$$\mathcal{L}_{DSL} = \sum_{i=1}^4 \overline{\psi}_i i \slashed{D}_a \psi_i + \frac{1}{4e^2} f_{\mu\nu} f^{\mu\nu}, \tag{3}$$

where the fermions are minimally coupled to the dynamical $U(1)$ gauge field $a$ with curvature $f_{\mu\nu} = \partial_\mu a_\nu - \partial_\nu a_\mu$.

## 2.1 Monopole operators and the IR global symmetry

The quantum field theory described by Eq. (3) has been studied extensively. The Dirac fermion $\psi$ is not a local (gauge invariant) observable but operators such as $\bar{\psi} O \psi$, where $O$ is hermitian, are local. A different class of local observables are monopole operators $\mathcal{M}_q$, which insert $q$ units of $U(1)$ gauge flux into the system. In the absence of such monopole operators in the action, the total flux of the $U(1)$ gauge field is conserved, corresponding to a global $U(1)$ symmetry (denoted $U(1)_{top}$). There is a corresponding conserved current

$$j^\mu = \frac{1}{2\pi} \epsilon^{\mu\nu\lambda} \partial_\nu a_\lambda. \tag{4}$$

The monopole operators $\mathcal{M}_q$ carry charge $q$ under $U(1)_{top}$. In the presence of Dirac fermions, the bare monopole operators $\mathcal{M}_q$ must be dressed by fermion zero modes in order to be gauge invariant [27]. Specifically, gauge invariance requires each bare monopole operator to have half of the 4 Dirac zero modes filled, so in total, there are 6 fundamental monopoles [27] arising from $\mathcal{M}_1$ and the 2 out of 4 choices of Dirac zero mode fillings.

All local operators, including fermion bilinears such as $\overline{\psi}_i O \psi_j$, can be constructed as composites of the gauge-invariant monopole operators. Thus the monopole operators may be viewed as the 'fundamental' degrees of freedom of the theory.

---

[2]We include a subscript $g$ to emphasize that this is a gauge group.

At first cut, apart from $U(1)_{top}$, the QED$_3$ Lagrangian has an $SU(4)_f$ flavor symmetry corresponding to rotations of the $N_f = 4$ $\psi$-fields. The monopoles transform as a vector under $SO(6) = SU(4)_f/\mathbb{Z}_2$ [17]. The correct internal symmetry group[3] is actually $G_{QED} = \frac{SO(6) \times U(1)_{top}}{\mathbb{Z}_2}$. In addition there are discrete symmetries of time reversal, reflection, and Lorentz symmetry.[4] A summary of the local operators and their symmetry transformations is collected in Appendix A.

## 2.2 Embedding the microscopic global symmetry

In any quantum many body system, the microscopic (the UV theory, in field theory parlance) system has a global symmetry group $G_{UV}$. This will be embedded in the global symmetry of the IR theory $G_{IR}$ through a homomorphism. For the lattice spin model, $G_{UV} = SO(3) \rtimes \mathcal{T} \times p4m$, where $p4m$ is the square lattice symmetry group, generated by the unit lattice translations $T_{1,2}$, $C_4$ rotation, and mirror reflection $R_x$. Time reversal $\mathcal{T}$ acts as an antiunitary symmetry that reverses monopole flux and fermion spin. A table of how these symmetries act on the fermion fields and the gauge fields can be found in Ref. [9], which then determine how gauge invariant operators like fermion bilinears transform under action of (the IR image of) $G_{UV}$. There is no such fermion bilinear that is a singlet under $G_{UV}$.

The transformation of the monopole operators is more subtle. It was found through analyzing the embedding of the spin liquid projective symmetry group into $G_{QED}$ that, of the 6 single-strength monopoles, there is one, $\Phi_{triv}$, whose imaginary part transforms trivially under $G_{UV}$ for the staggered flux/dRVB state on the square lattice, invariant under $G_{UV}$ [16,17,20]. The other 5 single strength monopoles and corresponding antimonopoles all transform under $G_{UV}$. The symmetry properties of other operators in the theory follow directly from those of the single strength monopoles and fermion bilinears and is summarized in Appendix A.

# 3 Critical theory of the DSL

In the absence of the massless Dirac fermions, any allowed monopole perturbation will gap out the photon field and confine gauge charges [28]. This is modified with the addition of $N_f \neq 0$. In the large $N_f$ limit, the monopole perturbation becomes irrelevant, and it is known that Eq. (3) is critical and flows to a conformal fixed point [7,29–34]. As $N_f$ is reduced, if monopole operators are not included, the QED$_3$ theory is believed to flow to a CFT (at least down to $N_f = 4$ of interest to us here). If now monopoles are included at this conformal fixed point and are relevant, the spin liquid phase is unstable.

The QED$_3$ theory has been studied through the large-$N_f$ expansion [5,6,9,27,34], and through direct Monte Carlo calculation [32,33,35]. In addition, conformal bootstrap calculations [36–38] have obtained constraints on the scaling dimensions of various operators, though a full isolation of the corresponding CFT has thus far not been possible.

From a $1/N_f$ expansion, the scaling dimension of the fundamental monopoles $\Phi$ is given by [27,34]

$$\Delta_1 = 0.265 N_f - 0.0383 + \mathcal{O}(1/N_f). \tag{5}$$

Extrapolating to $N_f = 4$ gives $\Delta_1 \sim 1.02 < 3$ which would be relevant. This expectation is confirmed by direct Monte Carlo calculations [35]. We can also consider composite operators obtained by taking a product of the single monopole and a fermion bilinear, *i.e* terms like

---

[3]Rotations by the $Z_2$ element of the center of $SO(6)$ can be compensated by a $U(1)_{top}$ rotation which is why $SO(6) \times U(1)_{top}$ is modded by $Z_2$.

[4]The Lorentz symmetry is obviously not present in the lattice model but is an emergent low energy symmetry, as can be shown explicitly within a large-$N_f$ expansion [9].

$\overline{\psi}_i \psi_j \Phi$. In the large-$N_f$ limit, the scaling dimension of this operator is $\Delta_1 + 2\sqrt{2}$[5] which extrapolates to $\sim 3.8 > 3$ for $N_f = 4$ making these irrelevant. The strength 2 monopole scaling dimension, estimated from large $N_f$, is $\Delta_2 = 0.673 N_f - 0.194 \sim 2.5 < 3$ and is nominally slightly relevant [34]. However, recent Monte Carlo calculations [35] estimate $\Delta_2 = 3.73(34)$ and comfortably favors the irrelevance of the strength 2 monopole operator. We will assume this below. Then the only relevant operator we need to consider is the fundamental strength one monopole, $\Phi$. If allowed by symmetry, proliferation of $\Phi$ will prevent the DSL from being a stable gapless phase.

For the staggered flux/dRVB state on the square lattice, there is a single strength-1 monopole allowed by the symmetries of the microscopic theory. Fermion bilinears are also relevant but are not singlets under $G_{UV}$. Included in the fermion bilinears are the Neel order parameter $\vec{N}$ (which transforms as spin-1 under $SO(3)$) and the VBS order parameter $\psi_{VBS}$ (which is a spin $SO(3)$ singlet).

To be thorough, we must consider operators that correspond to four fermion interactions and higher order monopoles, which may be relevant. These can be organized into various representations of $SO(6)$, including symmetry-allowed operators that transform in the symmetric tensor representation ($\mathbf{20'}$) and higher representations of $SO(6)$. The symmetric tensor representation includes the operator $\vec{N}^2 - |\psi_{VBS}|^2$, which will play an important role in our analysis below. It has been argued that this operator will be allowed in any lattice discretization of QED$_3$, and hence will be irrelevant since the CFT is found in a simulation [37]. Based on this evidence, we assume this operator is irrelevant.

Furthermore, we will assume that the UV symmetry allowed operators in the higher tensor representations of $SO(6)$ are irrelevant. These assumptions are similar to the ones needed for a stable spin liquid on the Kagome lattice [37] and are listed in Appendix A. With these assumptions, there will be precisely one relevant symmetry allowed operator, and the square lattice DSL is not a stable phase but rather a quantum critical point. Placing this quantum critical point in the phase diagram of the square lattice spin-1/2 magnet is the main goal of this paper.

Lastly, we recall, as mentioned earlier, that all fermion bilinears and higher-order fermion terms can be written in terms of monopole operators. In particular, the fermion billinears correspond to monopole-antimonople pairs while four fermion operators correspond to composites of higher order monopole-antimonopole pairs (Appendix A).

## 4 Effective theory of the perturbed DSL

We consider the continuum quantum field theory defined by perturbing the QED$_4$ theory with its single allowed relevant perturbation.

$$\mathcal{L} = \mathcal{L}_{DSL} + (\lambda i \Phi^\dagger_{triv} + h.c.) + \dots, \tag{6}$$

where the "…" includes symmetry-allowed (under $G_{UV}$) couplings which are irrelevant at the QED$_3$ fixed point. We will first analyse the properties of the continuum field theory defined by the QED$_3$ CFT with its single relevant deformation, and then reinstate these extra terms to connect to the phase diagram of the microscopic lattice spin system.

Note there are 6 monopoles and 6 antimonopoles and only a particular linear combination of them, $\text{Im}[\Phi_{triv}]$, is symmetry allowed. Once $\lambda \neq 0$, the presence of the trivial monopole

---

[5]The scaling dimension is derived from the state-operator correspondence. Such an operator corresponds to an excited $2\pi$ lorentz singlet monopole. The leading-order operator of this kind results from exciting a single sphere Landau level in a unit charge monopole background from level $n = -1$ to $n = 1$, which has excitation energy of $2\sqrt{2}$.

of course breaks the $U(1)_{top}$ symmetry. Further it reduces the flavor symmetry from $SO(6)$ to $SO(5)$, with the real and imaginary parts of $\Phi_{triv}$ being $SO(5)$ scalars. The remaining 10 linear combinations of monopoles and antimonopoles will split into two sets of $SO(5)$ vectors. There are symmetry allowed terms in the Lagrangian that couple one set of 5 to four-fermion terms and the remaining set of 5 to the adjoint fermion billinear masses after $\lambda \neq 0$. However, we will focus on the condensation of the particular set of 5 operators that couple to the adjoint fermion billinear mass[6] after $\lambda \neq 0$, as the fermion bilinears are more relevant than the four fermion terms at the $\text{QED}_3$ fixed point. We label these monopole operators as $n^a$ with $a = 1, ..., 5$.

The transformation of $n^a$ under $G_{UV}$ is such that we can identify them with the 5 component vector $\hat{n} = (N_x, N_y, N_z, \text{Re}\,\psi_{VBS}, \text{Im}\,\psi_{VBS})$, where $N_{x,y,z}$ is the Neel order parameter and $\psi_{VBS}$ is the columnar valence bond solid (VBS) order parameter [16,20]. Time reversal $Z_2^T$ continues to be a symmetry. We will take the corresponding operation $\mathcal{T}_{IR}$ to change the sign of all 5 components of $n^a$. The UV time-reversal operation $\mathcal{T}_{UV}$ maps in the IR theory to a combination of $\mathcal{T}_{IR}$ and an $SO(5)$ rotation that flips the sign of the VBS order parameters. UV reflection along, say, the $x$-axis acts as a unitary $Z_2$, reversing one of the $SO(5)$ vector components. As with time reversal, we can combine this with an $SO(5)$ rotation to define an IR reflection operation $R_x^{IR}$ which changes the sign of all 5-components of $n^a$.

The relevance of the single monopole means that a non-zero $\lambda$ will grow upon scaling to low energies, and will lead to an expectation value for $\Phi_{triv}$. The 5-component $n^a$ operator, representing the non-singlet monopoles, can also be viewed as fermion bilinears that transform as a vector under $SO(5)$ (in other words once $\langle\Phi_{triv}\rangle \neq 0$, there is no symmetry distinction between the remaining monopoles and these fermion bilinears). To be more precise, at $\lambda = 0$, the 15 fermion billinear masses transform as the **15** representation of $SO(6)$ which branches into **5** $\oplus$ **10** under $SO(6) \to SO(5)$. It is the **5** that can be identified with $n^a$ once $\lambda \neq 0$.

It was shown in Ref. [39] that there is an anomaly for the global $SO(5) \times Z_2^T$ symmetry. The part of the anomaly involving just the $SO(5)$ symmetry can be characterized physically as follows. Consider a vortex in two components of the $SO(5)$ vector $\hat{n}$. This vortex configuration breaks the $SO(5)$ to $SO(3) \times U(1)$. The anomaly implies that the vortex transforms as a spinor under the remaining $SO(3)$. This is simply the familiar fact [40] that VBS vortices for square lattice spin-1/2 magnets transform as spinors under the global $SO(3)$ spin rotation symmetry. We can also characterize the anomaly formally by coupling a background $SO(5)$ gauge field, and placing the theory on an arbitrary oriented smooth space-time manifold $X_3$. As usual, the anomaly implies that the action will not be invariant under $SO(5)$ gauge transforms. To get a gauge-invariant action, we need to extend the $SO(5)$ gauge field, but not the dynamical fields, to one higher space-time dimension $X_4$ whose boundary is $X_3$. The bulk action for the $SO(5)$ gauge field will be the response of an invertible topological phase such that the combined bulk-boundary action is gauge invariant. The anomaly can thus be fully characterized by specifying the response of the invertible topological phase on a closed compact manifold $M_4$. For the perturbed $N_f = 4$ $\text{QED}_3$ theory, this response takes the form of a discrete theta term

$$\mathcal{Z}_{bulk} = \exp\left(i\pi \int_{M_4} w_4^{SO(5)}\right), \tag{7}$$

where $w_4^{SO(5)} \in H^4(X_4, \mathbb{Z}_2)$ is the fourth Stiefel-Whitney class of the $SO(5)$ gauge bundle on $M_4$. The full anomaly (including time-reversal) has a similar characterization but with $w_4^{O(5)}$ where the improper element involves orientation reversal of the space-time manifold [41]. Crucially, this full anomaly is also $\mathbb{Z}_2$ classified, so it is independent of the sign of $\lambda$ and disappears if two copies of the theory are stacked.

---

[6]The specific form of the coupling can be found in [16].

Both the symmetry and the anomaly are identical to that describing the putative deconfined quantum critical point [26, 39] between Neel and VBS phases. This is perhaps not surprising given the identification of the basic uncondensed monoples of the the deformed $QED_3$ theory with the Neel and VBS order parameter fields. Any field theory describing the fluctuations of these orders must have an anomaly that reflects lattice Lieb-Schultz-Mattis constraints.

## 5 Symmetry breaking and relation to the DQCP

What is the fate of the deformed $QED_3$ theory with its anomalous $SO(5) \times Z_2^T$ symmetry? First we know that any anomaly prevents a trivial gapped phase that preserves the global symmetry. Ref. [39] argued a stronger result: so long as $SO(5) \times Z_2^T$ symmetry is preserved, even a gapped topological ordered phase is forbidden, *i.e*, the theory has the property [42] of 'symmetry-enforced gaplessness'.

The most likely fate of the deformed $QED_3$ theory is simply a state where the $SO(5)$ is spontaneously broken. To understand how this might come about, first consider the theory in the absence of any monopoles. Then the fermions of $QED_3$ can be viewed as the basic $2\pi$ vortices of an order parameter carrying charge-1 under $U(1)_{top}$. This arises from the reciprocal $2\pi$ Berry phase when a $U(1)_{top}$ charged object, e.g. an elementary monopole, is moved around a fermion in a closed loop. The addition of the singlet monopole operator leads to a field that couples linearly to this order parameter. The phase winding around vortices will then be restricted to strings that connect vortices to anti-vortices. Thus monopole proliferation leads to a strong attractive interaction between fermionic particles and their holes. Thus we expect a particle-hole (rather than a pairing) condensate to form once the monopole fugacity becomes large. A further clue on the nature of this condensate comes from examining the particle-hole operator with the lowest scaling dimension at the $QED_3$ fixed point: it is presumably these operators that will acquire an expectation value once the monopole fugacity flows to strong coupling. It is known from calculations of scaling dimensions [9, 10, 22] (from the large-$N_f$ expansion) that the flavor mass operator $\bar{\psi}M\psi$, where $M$ is an adjoint $SU(4)$ matrix, has a lower scaling dimension than the singlet mass operator $\bar{\psi}\psi$. Therefore, the dominant, slowly decaying, long-wavelength correlations at the $QED_3$ fixed point will arise from the flavor mass operator $\bar{\psi}M\psi$. Due to these enhanced correlations at the $QED_3$ fixed point, we heuristically expect flavor symmetry to be broken.[7] Further, as we discussed above, the true symmetry of the model in the presence of the singlet monopole is $SO(5)$, and we might reasonably expect that the condensate transforms in the vector representation of $SO(5)$.

A low energy continuum effective theory that captures this symmetry broken state is a sigma model in terms of the 5-component $SO(5)$ unit vector $\hat{n}$ with a level 1 Wess-Zumino-Witten (WZW) term [43, 44],

$$S = \frac{1}{2g} \int d^3x \, (\partial \hat{n})^2 + 2\pi \Gamma[\hat{n}], \qquad (8)$$

where the WZW term $\Gamma$ is defined by extending $\hat{n}$ into an extra dimension $u$. This is done by

---

[7]The intuition is that the operator with the slower correlations at the CFT fixed point is more "almost ordered" and if there is a relevant perturbation, then it is more likely to freeze in the fluctuations of this operator, though a more thorough analysis would require a consideration of the parameter space surrounding the fixed point. A more concrete example of this heuristic argument is to consider an array of spin-1/2 chains coupled together by antiferromagnetic interactions between nearest neighbor chains. Each spin chain has power law Neel and VBS correlations; the Neel correlations are enhanced over the VBS by a log factor, so the Neel is (slightly) more slowly fluctuating than the VBS. Now as the inter-chain coupling is known to be relevant, the belief is that the relevant flow leads to the Neel ordered state, rather than the VBS ordered state (at least so long as the interchain interaction is not frustrating).

defining $\widehat{n}(\boldsymbol{x}, u)$ as a smooth extension of $\widehat{n}(\boldsymbol{x})$ with $\widehat{n}(\boldsymbol{x}, u = 0) = \widehat{n}(\boldsymbol{x})$ and $\widehat{n}(\boldsymbol{x}, u = 1) = \widehat{n}^{ref}$ a fixed reference vector (for example, $\widehat{n}^{ref} = (1, 0, 0, 0, 0)^T$). In terms of the extended vector $\widehat{n}$, the WZW term is written as

$$\Gamma[\widehat{n}] = \frac{\epsilon_{abcde}}{Vol(S^4)} \int_0^1 du \int d^3x \, n^a \partial_x n^b \partial_y n^c \partial_y n^d \partial_u n^e. \tag{9}$$

The WZW term endows the sigma model with the right anomaly for the $SO(5) \times Z_2^T$ symmetry. As a continuum theory, the sigma model is well defined at weak coupling and flows to the ordered fixed point where $SO(5)$ is spontaneously broken to $SO(4)$. That the sigma model matches the correct anomaly of the deformed QED$_3$ is further evidence supporting an $SO(5)$ ordered ground state.

In principle, this theory could flow to a gapped, chiral spin liquid that saturates the $SO(5)$ anomaly [39]. However, we rule out this possibility[8] as the resulting topological order has an intrinsic sign problem [45], while the deformed QED$_3$ is sign-problem free, both on the lattice [46] and in the continuum field theory.

We comment briefly on the more exotic possibility that the IR fixed point of the deformed QED$_3$ theory is an interacting gapless CFT with the anomalous global $SO(5)$ symmetry. This fixed point needs to have no relevant $SO(5) \times Z_2^T$ invariant perturbations so that the perturbed QED$_3$ theory can flow to it without fine tuning. If breaking $SO(5)$ to $SO(3) \times SO(2)$ is relevant, such a theory would describe the deconfined quantum critical point between Neel and VBS states (with emergent $SO(5)$ symmetry), does not seem to exist as per numerical calculations on lattice models, a conclusion supported by searches using the conformal bootstrap [47,48]. Rather the Neel-VBS transition seems to be very weakly first order which corresponds, in the $SO(5)$ symmetric model, to a weak breaking of the $SO(5)$ symmetry.[9] An even more exotic possibility is that such a gapless CFT exists with no relevant perturbations even if $SO(5)$ is broken to $SO(3) \times SO(2)$. Such a CFT might describe a stable gapless phase of the underlying lattice spin system. In the absence of any strong evidence supporting the existence of such a phase, we do not consider it further here.[10]

## 5.1 DSL as an unnecessary critical point

The DSL quantum critical point is obtained by tuning the sign of the monopole fugacity $\lambda$. The theory has the same symmetry and anomaly for either sign of $\lambda$. Moreover, the sign of $\lambda$ can be flipped by an $SO(6)$ rotation (i.e. the center of the group) in the DSL. Thus irrespective of the sign of $\lambda$ we expect that it flows to the $SO(5)$ symmetry broken ground state where the $\widehat{n}$ condenses. It follows that the QED$_3$ CFT at $\lambda = 0$ is a quantum critical point within the $SO(5)$ symmetry broken ordered state.

When the perturbed QED$_3$ arises from microscopic lattice models, it will have anisotropies that break $SO(5)$ to a smaller subgroup. An important such anisotropy is the term

$$\mathcal{L}_1 = -\kappa \left[ 2 \left( n_1^2 + n_2^2 + n_3^2 \right) - 3 \left( n_4^2 + n_5^2 \right) \right], \tag{10}$$

which breaks $SO(5)$ to $SO(3) \times U(1)$, and selects between Neel and VBS orderings. This perturbation transforms in the traceless symmetric tensor representation, $\mathbf{20'}$, of $SO(6)$ and is

---

[8]We thank Chong Wang for pointing this out to us.

[9]It is possible that an $SO(5)$ symmetric CFT exists as a complex fixed point [39, 49–52] that leads to a slow walking renormalization group flow to the symmetry broken fixed point, or that the weakness of the ordering is due to proximity of the studied models to a tricritical point [53–55]. But whatever the explanation, $SO(5)$ symmetry breaking, rather than conformality, seems to be the ultimate fate.

[10]A simple symmetry preserving gapless phase is one where the Dirac fermions pair condense, thereby Higgsing the $U(1)$ gauge field to $Z_2$. However we alreay argued against such pair condensation when the QED$_3$ theory is destabilized by monopoles.

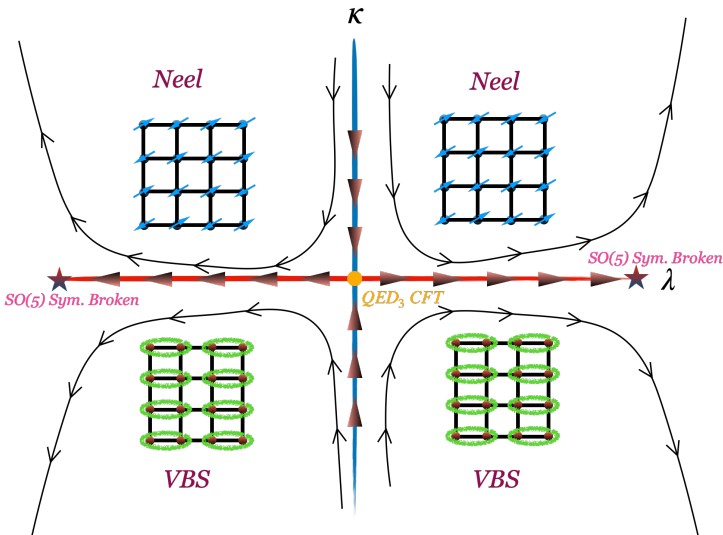

Figure 2: A phase diagram with RG flow of the unnecessary QCP. The horizontal axis is the monopole fugacity $\lambda$, which drives the QED$_3$ CFT into the $SO(5) \to SO(4)$ symmetry broken state. The vertical axis is a generic, dangerously irrelevant anisotropy that tunes between Neel and VBS ordering.

hence expected to be irrelevant at the QED$_3$ fixed point. When a non-zero $\lambda$ is turned on, the sign of $\kappa$ nevertheless determines whether the $SO(5)$ broken state will possess Neel or VBS order. Thus $\kappa$ should be viewed as 'dangerously irrelevant'. In general, different microscopic lattice models will yield different signs of $\kappa$. Thus, in a class of microscopic Hamiltonians, we expect to find the QED$_3$ CFT as a quantum critical point inside the Neel phase (and likewise also find it as a quantum critical point inside the VBS phase for other microscopic models).

As $\kappa$ is dangerously irrelevant and $\lambda$ is relevant, we can obtain the structure of the renormalization group (RG) flows in the $(\kappa, \lambda)$-plane is shown in Figure 2, with a schematic phase diagram as well. More generally, the unnecessary critical point will occupy some codimension one surface in parameter space.

As opposed to QCPs that separate two different phases, we see the DSL QCP is an *unnecessary* QCP, as it describes a continuous transition within the same phase. In particular, we can find the DSL embedded within just the familiar collinear Neel phase, and hence may not require a too strongly frustrated Hamiltonian. These results are summarized in Figure 1.

Implicitly, in reaching the conclusions above, we assumed that the sign of $\kappa$ cannot change under the RG flow when the monopole fugacity $\lambda$ flows to strong coupling. In particular, if the sign of the renormalized $\kappa$ was determined by the sign of $\lambda$ (irrespective of the bare sign of $\kappa$), then we will have a situation where for one sign of $\lambda$ we end up in the Neel state, and for the other sign of $\lambda$, in the VBS phase. The QED$_3$ CFT will then describe a second order Neel-VBS transition. However this possibility is disallowed on general grounds simply because of $SO(5)$ symmetry which dictates that $d\kappa/dl$ must vanish at $\kappa = 0$. Therefore, the flows of $(\kappa, \lambda)$ cannot cross the $\kappa = 0$ axis.[11]

Since the phases on either side of the unnecessary QCP are the same, we expect that there will be a smooth path that connects them which avoids the QED$_3$ completely. Thus the unnecessary phase boundary will end within the Neel phase at a multicritical point, which may also be interesting to explore in future work. An obvious candidate (within the Neel phase)

---

[11]Here we make the further assumption that other $SO(5)$ breaking perturbations with higher scaling dimension (at the QED$_3$ fixed point) generated once $\kappa \neq 0$ stay sufficiently small that we can ignore them and study the theory in the $(\kappa, \lambda)$ plane; then the $\kappa = 0$ theory is $SO(5)$ invariant.

is the QED$_3$-Gross-Neveu model studied in Ref. [56]. Here the QED$_3$ action is modified by including a coupling to a fluctuating Neel vector field $\vec{\phi}$ that couples to a fermion bilinear with the same transformation under $G_{UV}$. The Lagrangian reads

$$\mathcal{L}_{QED-GN} = \sum_{i=1}^{4} \overline{\psi}_i i \not{D}_a \psi_i + \vec{\phi} \cdot \bar{\psi} \mu^z \vec{\sigma} \psi + \left( \partial_\mu \vec{\phi} \right)^2 + r \vec{\phi}^2 + \cdots + \frac{1}{4e^2} f_{\mu\nu} f^{\mu\nu}. \tag{11}$$

(Here $\mu^I$ are Pauli matrices in the Dirac valley space, and $\vec{\sigma}$ are Pauli matrices in physical spin space.) When $\vec{\phi}$ is gapped, we can integrate it out to get the QED$_3$ theory that describes the unnecessary critical line. When $\vec{\phi}$ is condensed, there is Neel order and the fermions get a mass gap. The gauge fluctuations and the fermions will then be confined, and we get the conventional Neel state. More detail is in Ref. [56]. A similar theory with a fluctuating VBS order parameter field replacing the Neel field is also a candidate for the multicritical point inside the VBS phase.

Finally we mention that there are other theories that share the same $SO(5)$ global symmetry and anomaly such as $N_f = 2$ QCD$_3$, of which the DSL can be realized as a Higgs descendant [39]. The picture of the DSL as a parent state of the DQCP can also be seen from the theory of Stiefel liquids as formulated in Ref. [41]. The $SO(6)/SO(2)$ Stiefel liquid with a WZW term at level 1 emerges as a theory closely related to the QED$_3$ CFT, in the sense of having the same local operators, global symmetry, and anomaly. Condensing a single monopole is equivalent to condensing part of the Stiefel manifold order parameter, resulting in nothing more than the $SO(5)$ NLSM in Eq. (8).

# 6 Experiments and discussion

In addition to providing a platform to realize competing orders on different lattices, the DSL can be realized as a critical theory living inside a single ordered phase on the square lattice. Experimentally, the proximity to the DSL QCP within the Neel phase could be probed through scattering experiments. A Neel state proximate to the unnecessary QCP would still exhibit conventional behavior at long wavelengths and lowest energy, exhibiting magnon modes from the spontaneously broken spin symmetry. However at higher energy, one would be able to probe decay of the magnon into the CFT modes, leading to behavior such as anomalous broadening of the magnon spectral function. The magnitude of the Neel order parameter, as well as the spin stiffness, will itself vanish upon approaching the DSL QCP with exponents related to the scaling dimension of the fermion billinears, as shown in Figure 1 b). Similarly, when the DSL QCP arises as an unnecessary QCP within the VBS state, there will be vanishing of both the order parameter and the energy gap as one tunes toward the DSL.

The example we have presented in this paper allows for unexpected phenomenology even in systems deep within conventional phases of matter. Clearly it would be interesting to search for "unnecessary" quantum criticality inside the Neel phase through numerical simulations of square lattice spin-1/2 magnets.

It is also interesting to note the implications for theories of superconductivity in doped Mott insulators. The DSL we have studied has an approximate wavefunction description as a Gutzwiller-projected nearest neighbor d-wave BCS superconductor. At half-filling this is a (spin liquid) Mott insulator. The same wavefunction at non-zero doping describes a correlated d-wave superconductor. Thus the doped DSL gives us a route to connect the Mott insulator and the doped superconductor, as explored in the old literature [19]. The traditional view has been that the parent DSL state of the doped superconductor can only be accessed in a deformation of the realistic spin Hamiltonian that is strong enough to completely leave the Neel state. The understanding developed here shows that this parent state exists already within the Neel state, and thus the deformation needed to access it may not be as violent as previously assumed.

## Acknowledgments

We thank Shai Chester, Yin-Chen He, Ho Tat Lam, Max Metlitski, Silviu Pufu, Chong Wang and Xiao Yan Xu for useful discussions.

**Funding information**  YZ was supported by the National Science Foundation Graduate Research Fellowship under Grant No. 2141064. TS was supported by NSF grant DMR-2206305, and partially through a Simons Investigator Award from the Simons Foundation. XYS was supported by the Gordon and Betty Moore Foundation EPiQS Initiative through Grant No. GBMF8684 at the Massachusetts Institute of Technology. This work was also partly supported by the Simons Collaboration on Ultra-Quantum Matter, which is a grant from the Simons Foundation (Grant No. 651446, T.S.). This research was supported in part by grant NSF PHY-1748958 to the Kavli Institute for Theoretical Physics (KITP).

## A   UV symmetry allowed operators in QED$_3$

In this appendix, we will analyze the symmetry properties of higher monopole operators in QED$_3$. Specifically, we will consider composites of the fundamental monopoles and categorize them into representations of the global IR $SO(6)$ and $U(1)_{top}$ symmetry. If the DSL is to be a quantum critical point and not a multicritical point, then except for the operator used to tune the critical point, all other operators allowed under the UV symmetries should be assumed to be irrelevant. We will relate these assumptions to statements about the stability of DSLs on the triangular and Kagome lattice made in Ref. [37].

The transformation of the single charge monopoles under the UV symmetries was derived in Ref. [17] and is outlined in Table 1.

We now describe the symmetry allowed operators in each sector. We will label the representations by $(\mathbf{d}, q)$ where $\mathbf{d} = \dim(rep(SO(6)))$ and $q$ the $U(1)_{top}$ charge. We remark that all allowed operators of the form $(\mathbf{d}, q)$ will appear in the theory as linear combinations of $(\mathbf{d}, \pm q)$ (or equivalently, the real and imaginary parts of $(\mathbf{d}, q)$). Here are the assumptions made about the operators for DSL stability on the triangular and Kagome lattices, in which there is no symmetry allowed single strength monopole:

- Triangular lattice: We only need to assume $(\mathbf{20'}, 0)$ and $(\mathbf{84}, 0)$ are irrelevant, as only strength 3 monopoles are allowed.

- Kagome lattice: We must assume $(\mathbf{20'}, 0)$, $(\mathbf{84}, 0)$, $(\mathbf{20'}, 2)$, and $(\mathbf{64}, 1)$ are irrelevant.

Using results from Refs. [16,17], Ref. [37] derived the above UV singlet operators that must be RG irrelevant in order for the DSL to be stable. We note the operators $(\mathbf{1}, 0)$ must be irrelevant in order for the QED$_3$ to flow to a true CFT in the IR without fine tuning. Furthermore, $(\mathbf{20'}, 0)$

Table 1: The transformation of the single-charge monopoles under the UV symmetries. The trivial monopole is $\mathrm{Im}[\Phi_2]$.

| Monopole | $T_1$ | $T_2$ | $R_x$ | Rotation | $\mathcal{T}$ |
|---|---|---|---|---|---|
| $\Phi_1^\dagger$ | $\Phi_1$ | $-\Phi_1$ | $-\Phi_1$ | $-\Phi_3$ | $\Phi_1^\dagger$ |
| $\Phi_2^\dagger$ | $-\Phi_2$ | $-\Phi_2$ | $-\Phi_2$ | $-\Phi_2$ | $-\Phi_2^\dagger$ |
| $\Phi_3^\dagger$ | $-\Phi_3$ | $\Phi_3$ | $\Phi_3$ | $\Phi_1$ | $\Phi_3^\dagger$ |
| $\Phi_{4/5/6}^\dagger$ | $-\Phi_{4/5/6}$ | $-\Phi_{4/5/6}$ | $\Phi_{4/5/6}$ | $\Phi_{4/5/6}$ | $-\Phi_{4/5/6}^\dagger$ |

Table 2: Representations of $q = 0$.

| Rep. | Tensor form | UV Symmetry Transformation |
|---|---|---|
| $(\mathbf{1}, 0)$ | $\sum_i \mathcal{O}_i^\dagger \mathcal{O}_i$ | Allowed |
| $(\mathbf{15}, 0)$ | $\text{Im}[\mathcal{O}_i^\dagger \mathcal{O}_j]$ | Transforms as the fermion billinears |
| $(\mathbf{20'}, 0)$ | $\mathcal{O}_{(i}^\dagger \mathcal{O}_{j)} - \delta_{ij} \frac{\sum_k \mathcal{O}_k^\dagger \mathcal{O}_k}{6}$ | Allowed, such as $(\mathcal{O}_1^\dagger \mathcal{O}_1 + \mathcal{O}_3^\dagger \mathcal{O}_3 - \cdots)$ |

and $(\mathbf{84}, 0)$ contain singlets in any lattice QED$_3$ simulation and therefore must be irrelevant if $N_f = 4$ QED$_3$ is found to be stable in any lattice gauge model.

Now let us make the same assumptions for the square lattice as Ref. [37] does for the stability of the Kagome lattice DSL and observe if any additional conditions are required in order for the square lattice DSL to not be a multicritical point. For ease of notation, we write $\mathcal{O}_{i_1,\dots i_n} = \Phi_{i_1} \dots \Phi_{i_n} = \mathcal{O}_{(i_1,\dots i_n)}$. In each $U(1)_{top}$ charge sector, we will first describe the irreducible representations at fixed $q$ and then comment on what is UV allowed. As $\mathcal{O}_\bullet$ transforms as the single index vector representation of $SO(6)$, we can analyze the composites of $\mathcal{O}$ to derive explicit tensor forms of each representation and examples of symmetry allowed operators.

Viewing the $SO(6)$ fundamental representation as the antisymmetric two box representation of $SU(4)$, we can label the $\mathbf{6}$ of $SO(6)$ as a Young tableau with a single column of two boxes. Then, in general, for a charge $q$ operator, its allowed $SO(6)$ representations will labeled by $SU(4)$ Young tableau with $\nu$ boxes, where $\frac{\nu}{2} = q$ (mod 2).

## A.1 $q = 0$

In the $q = 0$ sector, we have operators of the form $\mathcal{O}_\bullet^\dagger \mathcal{O}_\bullet, \mathcal{O}_{\bullet\bullet}^\dagger \mathcal{O}_{\bullet\bullet}, \cdots$. The first term splits as

$$\mathbf{6} \otimes \mathbf{6} = \mathbf{1} \oplus \mathbf{15} \oplus \mathbf{20'}, \tag{A.1}$$

under $SO(6)$, with explicit tensor forms and examples of symmetry allowed operators shown in Table 2, while the latter term splits as

$$(\mathbf{1} \oplus \mathbf{20'}) \otimes (\mathbf{1} \oplus \mathbf{20'}) = 2(\mathbf{1}) \oplus \mathbf{15} \oplus 3(\mathbf{20'}) \oplus \mathbf{84} \oplus \mathbf{105} \oplus \mathbf{175}, \tag{A.2}$$

as in Table 3. Note the $(\mathbf{105}, 0)$ is the fully symmetric, traceless representation, while $(\mathbf{84}, 0)$ contains all the rest of the operators symmetric with respect to $\mathcal{O} \to \mathcal{O}^\dagger$, minus the fully symmetric and trace components. We see the UV allowed operators lie in $(\mathbf{1}, 0)$, $(\mathbf{20'}, 0)$, and $(\mathbf{84}, 0)$, so these must be assumed irrelevant.

We note that fermion bilinears correspond to the operators transforming as $(\mathbf{15}, 0)$ in Table 2. The four-fermion term will have overlap with numerous sectors, including $(\mathbf{15}, 0)$, $(\mathbf{20'}, 0)$, and $(\mathbf{84}, 0)$ in Table 3, in addition to the $SO(6)$ representations $\mathbf{45}$ and $\overline{\mathbf{45}}$ that are hosted by higher order monopole-antimonopole operators.

## A.2 $q = 1, 3$

The $q = 1$ sector contains the operators of the form $\mathcal{O}_\bullet^\dagger, \mathcal{O}_{\bullet\bullet}^\dagger \mathcal{O}_\bullet, \cdots$. In the single monopole sector, we have the UV allowed trivial monopole, transforming in $(\mathbf{6}, 1)$. More specifically, it will be a linear combination of $(\mathbf{6}, 1)$ and $(\mathbf{6}, -1)$, $\text{Im}[\mathcal{O}_2^\dagger]$. The higher monopole sector carries representations

$$(\mathbf{1} \oplus \mathbf{20'}) \otimes \mathbf{6} = 2(\mathbf{6}) \oplus \mathbf{50} \oplus \mathbf{64}, \tag{A.3}$$

as described in Table 4. We see that the operators $(\mathbf{6}, 1)$, $(\mathbf{50}, 1)$, and $(\mathbf{64}, 1)$ are allowed under the UV symmetries. The symmetry channel $(\mathbf{6}, 1)$ is exactly the relevant tuning parameter. Therefore, we must assume $(\mathbf{50}, 1)$ and $(\mathbf{64}, 1)$ are irrelevant.

Table 3: More representations of $q = 0$.

| Rep. | Tensor form | UV Symmetry Transformation |
|---|---|---|
| $2(\mathbf{1}, 0)$ | $\sum_{i,j} \mathcal{O}_{ij}^\dagger \mathcal{O}_{ij}$<br>$\sum_{i,j} \mathcal{O}_{ii}^\dagger \mathcal{O}_{jj}$ | Allowed |
| $3(\mathbf{20'}, 0)$ | $\sum_i (\mathrm{Re}[\mathcal{O}_{ij}^\dagger \mathcal{O}_{ik}] - \delta_{jk} \frac{\sum_l \mathrm{Re}[\mathcal{O}_{ii}^\dagger \mathcal{O}_{il}]}{6})$,<br>$\sum_i (\mathrm{Re}[\mathcal{O}_{ii}^\dagger \mathcal{O}_{jk}] - \delta_{jk} \frac{\sum_l \mathrm{Re}[\mathcal{O}_{ii}^\dagger \mathcal{O}_{ll}]}{6})$,<br>$\sum_i (\mathrm{Im}[\mathcal{O}_{ii}^\dagger \mathcal{O}_{jk}] - \delta_{jk} \frac{\sum_l \mathrm{Im}[\mathcal{O}_{ii}^\dagger \mathcal{O}_{ll}]}{6})$ | Allowed |
| $(\mathbf{105}, 0)$ | $\mathcal{O}_{(ij}^\dagger \mathcal{O}_{kl)} - \frac{1}{10} \sum_m \big( \delta_{kl} \mathcal{O}_{(ij}^\dagger \mathcal{O}_{mm)} + \delta_{jl} \mathcal{O}_{(im}^\dagger \mathcal{O}_{km)} +$<br>$\delta_{il} \mathcal{O}_{(mj}^\dagger \mathcal{O}_{km)} + \delta_{jk} \mathcal{O}_{(im}^\dagger \mathcal{O}_{ml)} + \delta_{ik} \mathcal{O}_{(mj}^\dagger \mathcal{O}_{ml)} + \delta_{ij} \mathcal{O}_{(mm}^\dagger \mathcal{O}_{kl)} \big)$<br>$+ \frac{1}{80} \sum_{m,n} \big( \delta_{ij} \delta_{kl} \mathcal{O}_{(mm}^\dagger \mathcal{O}_{nn)} + \delta_{ik} \delta_{jl} \mathcal{O}_{(mn}^\dagger \mathcal{O}_{mm)} + \delta_{il} \delta_{jk} \mathcal{O}_{(mn}^\dagger \mathcal{O}_{nm)} \big)$ | Allowed, such as $(\mathcal{O}_{44}^\dagger \mathcal{O}_{44} - \cdots)$ |
| $(\mathbf{84}, 0)$ | Real part of $\mathcal{O}_{ij}^\dagger \mathcal{O}_{kl} - \mathcal{O}_{(ij}^\dagger \mathcal{O}_{kl)} - \frac{1}{6} \sum_m \big( \delta_{ij} (\mathcal{O}_{mm}^\dagger \mathcal{O}_{kl} - \mathcal{O}_{mk}^\dagger \mathcal{O}_{ml}) +$<br>$\delta_{kl} (\mathcal{O}_{ij}^\dagger \mathcal{O}_{mm} - \mathcal{O}_{mi}^\dagger \mathcal{O}_{mj}) \big) - \frac{1}{12} \sum_m \big( \delta_{ik} (\mathcal{O}_{mj}^\dagger \mathcal{O}_{ml} - \mathcal{O}_{mm}^\dagger \mathcal{O}_{jl}) + \delta_{il} (\mathcal{O}_{mj}^\dagger \mathcal{O}_{km}$<br>$- \mathcal{O}_{mm}^\dagger \mathcal{O}_{jk}) + \delta_{jk} (\mathcal{O}_{im}^\dagger \mathcal{O}_{ml} - \mathcal{O}_{mm}^\dagger \mathcal{O}_{il}) + \delta_{jl} (\mathcal{O}_{im}^\dagger \mathcal{O}_{km} - \mathcal{O}_{mm}^\dagger \mathcal{O}_{ik}) \big)$<br>$+ \big( \frac{1}{30} \delta_{ij} \delta_{kl} - \frac{1}{60} \delta_{ik} \delta_{jl} - \frac{1}{60} \delta_{il} \delta_{jk} \big) \sum_{m,n} (\mathcal{O}_{mm}^\dagger \mathcal{O}_{nn} - \mathcal{O}_{mn}^\dagger \mathcal{O}_{mn})$ | Allowed, such as $(\mathrm{Re}[\mathcal{O}_{44}^\dagger \mathcal{O}_{55}] - \cdots)$ |
| $(\mathbf{15}, 0)$ | $\sum_i \mathrm{Im}[\mathcal{O}_{ij}^\dagger \mathcal{O}_{ik}]$ | Transforms under symmetries |
| $(\mathbf{175}, 0)$ | Imaginary part of $\mathcal{O}_{ij}^\dagger \mathcal{O}_{kl} - \frac{1}{6} \sum_m \big( \delta_{ij} \mathcal{O}_{mm}^\dagger \mathcal{O}_{kl} + \delta_{kl} \mathcal{O}_{ij}^\dagger \mathcal{O}_{mm} \big) -$<br>$\frac{1}{8} \sum_m \big( \delta_{ik} \mathcal{O}_{mj}^\dagger \mathcal{O}_{ml} + \delta_{il} \mathcal{O}_{mj}^\dagger \mathcal{O}_{km} + \delta_{jk} \mathcal{O}_{im}^\dagger \mathcal{O}_{ml} + \delta_{jl} \mathcal{O}_{im}^\dagger \mathcal{O}_{km} \big)$ | Transforms under symmetries |

Table 4: Representations of $q = 1$.

| Rep. | Tensor form | UV Symmetry Transformation |
|---|---|---|
| $2(\mathbf{6}, 1)$ | $\sum_i \mathcal{O}_{ij}^\dagger \mathcal{O}_i$<br>$\sum_i \mathcal{O}_{ii}^\dagger \mathcal{O}_j$ | Allowed, such as $(\sum_j \mathrm{Im}[\mathcal{O}_{j2}^\dagger \mathcal{O}_j]$,<br>$\sum_j \mathrm{Im}[\mathcal{O}_{jj}^\dagger \mathcal{O}_2])$ |
| $(\mathbf{50}, 1)$ | $\mathcal{O}_{(ij}^\dagger \mathcal{O}_{k)} - \frac{1}{8} \sum_m \big( \delta_{ij} \mathcal{O}_{(mm}^\dagger \mathcal{O}_{k)} + \delta_{ik} \mathcal{O}_{(mj}^\dagger \mathcal{O}_{m)} + \delta_{jk} \mathcal{O}_{(im}^\dagger \mathcal{O}_{m)} \big)$ | Allowed, such as $(\mathrm{Im}[\mathcal{O}_{22}^\dagger \mathcal{O}_2] - \cdots)$ |
| $(\mathbf{64}, 1)$ | $\mathcal{O}_{ij}^\dagger \mathcal{O}_k - \mathcal{O}_{(ij}^\dagger \mathcal{O}_{k)} - \frac{2}{15} \delta_{ij} \sum_m (\mathcal{O}_{mm}^\dagger \mathcal{O}_k - \mathcal{O}_{mk}^\dagger \mathcal{O}_m) -$<br>$\frac{1}{15} \sum_m \big( \delta_{ik} (\mathcal{O}_{mj}^\dagger \mathcal{O}_m - \mathcal{O}_{mm}^\dagger \mathcal{O}_j) + \delta_{jk} (\mathcal{O}_{im}^\dagger \mathcal{O}_m - \mathcal{O}_{mm}^\dagger \mathcal{O}_i) \big)$ | Allowed, such as $(\mathcal{O}_4^\dagger \mathcal{O}_4 (\mathrm{Im}[\mathcal{O}_2]) - \cdots)$ |

For $q = 3$, we have operators such as $\mathcal{O}_{\bullet\bullet\bullet}^\dagger$, which transform as the fully symmetric

$$\mathbf{6} \oplus \mathbf{50}. \tag{A.4}$$

The $\mathbf{6}$ is the singlet $\sum_i \mathcal{O}_{iij}^\dagger$ while the $\mathbf{50}$ is the traceless part of $\mathcal{O}_{ijk}^\dagger$. Both of these sectors will contain UV symmetric operators, including $\sum_i \mathrm{Im}[\mathcal{O}_{ii2}]$ for the former and $\mathrm{Im}[\mathcal{O}_{222}]$ for the latter. However, as they have higher $U(1)_{top}$ charge $q = 3$, they are likely irrelevant and we will not consider them further.

### A.3  $q = 2$

The $q = 2$ sector contains operators of the form $\mathcal{O}_{\bullet\bullet}^\dagger$. These will transform in the symmetric

$$\mathbf{1} \otimes \mathbf{20'} \tag{A.5}$$

representation, as summarized in Table 5. As addressed in the main text, we have already assumed that the $q = 2$ monopole operators are irrelevant.

In conclusion, in addition to higher charge monopoles, we must assume the following operators irrelevant in order for the square lattice DSL to be a QCP:

- $(\mathbf{1}, 0)$: This has scaling dimension $\sim 4.23$ from large $N_f$ [57] and as mentioned previously, must be irrelevant to have a true CFT in the IR.

- $(\mathbf{20'}, 0)$: The lowest scalars in this channel are four-fermion operators with scaling dimensions $\sim 2.38$ from large $N_f$ [57].

- $(\mathbf{84}, 0)$: The lowest scalars are four-fermion operators with scaling dimension $\sim 4.54$ from large $N_f$ [57].

- $(\mathbf{50}, 1), (\mathbf{64}, 1)$: The lowest scalar in these channel is the excited $q = 1$ monopole, whose lowest dimension operator has scaling dimension $\sim 3.85$ by large $N_f$ [34].

These are the exact same assumptions made in [37] for the stability of the Kagome lattice DSL, with the addition of $(\mathbf{50}, 1)$ being irrelevant. Except for $(\mathbf{20'}, 0)$, all of these operators are estimated to be irrelevant from large $N_f$ calculations.

Table 5: Representations of $q = 2$.

| Rep. | Tensor form | UV Symmetry Transformation |
|---|---|---|
| $(\mathbf{1}, 2)$ | $\sum_i \mathcal{O}_{ii}^\dagger$ | Allowed |
| $(\mathbf{20'}, 2)$ | $\mathcal{O}_{ij}^\dagger - \delta_{ij} \frac{\sum_k \mathcal{O}_{kk}^\dagger}{6}$ | Allowed, such as $(\mathrm{Im}[\mathcal{O}_{44}^\dagger] - \cdots)$ |

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
