# Peer review of "Dirac spin liquid as an "unnecessary" quantum critical point on square lattice antiferromagnets"

_SciPost Physics Core, doi:SciPost Phys. Core 8, 024 (2025)_

## Round 2 · Referee Report · Anonymous (Referee 1) · 2024-7-21

Report

This paper deals argues that QED$_3$ with $N_f=4$ fermions can be realized as an "unnecessary" critical point sitting within the Neel phase of 2+1 D quantum antiferromagnets on the square lattice (and also within the VBS phase). "Unnecessary" here means that on both sides of the codimension one critical manifold we find the same phase. Their argument starts by identifying the relevant operator which has to be tuned at criticality (charge-1 monopole). Then they argue that for any size of the perturbing monopole coupling the RG flow terminates in the same theory, SO(5) sigma model with a WZW term, Eq. (8). Then they discuss how this picture changes in presence of (dangerously) irrelevant perturbations arising from the microscopic symmetry.

This is an interesting paper and I think it should be published. I have however a few questions and requests.

  • Consider the RG flows (6) with the irrelevant terms ... set to zero. Is the RG flow with $\lambda$ positive and negative supposed to be identical for all distances or only the IR fixed point (8) is supposed to be identical? The first possibility would be realized e.g. when perturbing the Ising model by the magnetic field perturbation, as is trivial to show since the sign of the coupling is flipped by a Z2. The second possibility is much more nontrivial. Whichever it is, it would be worth pointing out explicitly.

  • p.3 last but one paragraph "hence must be irrelevant". "Must be" sounds confusing here. Can the authors rephrase it more explicitly, e.g. that they believe it to be irrelevant based on this evidence and you will assume it so? (This is in line with the description of $\Delta_2$ above)

  • p.4 second paragraph. "One set of 5 will couple to four fermion terms" "orthogonal set of 5 operators that couple to the adjoint" I don't understand what the authors mean by these phrases. Could be some jargon that I'm not aware of. I'd be grateful for more details here.

  • p.5 first paragraph "has slower correlations" Can the authors explain what "slower" means here and why this implies the expectation that flavor symmetry is broken?

  • second column "supported by searches using the conformal bootstrap". Here and elsewhere the authors demonstrate familiarity with the conformal bootstrap results, which is great. Can they provide some references here, and they do in other places of the paper?

  • p.6 sexond->second

  • Appendix A. Are results in tables I,II,III,IV,V new? A citation or , alternatively, at least some details on the derivation would be needed.

  • Ref. [52] - "to a tricritical point" Several recent papers providing evidence for tricritical point include 2405.06607, 2405.04470, 2307.05307. I believe a reference these and other relevant works here would be appropriate, to properly balance the cited evidence for the complexed fixed points.

Recommendation

Ask for minor revision

---

## Round 2 · Referee Report · Anonymous (Referee 2) · 2024-7-24

Report

The manuscript proposed Dirac spin liquid (DSL) as an unnecessary quantum critical point in square lattice antiferromagnets. The main idea is based on the conjecture that, deforming DSL with the relevant $2\pi$ monopole will trigger a RG flow towards the SO(5) DQCP, which will further flow to an ordered phase due to some dangerously irrelevant operator. Therefore, DSL can in principle appear as a critical point, inside the same ordered phase, i.e. Neel order or valence bond solid.

The proposal is interesting, and would be helpful for future experimental or numerical search of DSL inside the more conventional ordered phase. I am happy to recommend this paper.

Requested changes

  1. In the second paragraph of Sec. IV, the authors wrote "Then the fermions of QED3 can be viewed as the basic $2\pi$ vortices...". I am not sure I understand this statement, maybe it is good to elaborate it a bit.

  2. Page 6, first paragraph, there is a typo, "The QED3 CFT will then describe a sexond..." should be second.

Recommendation

Publish (easily meets expectations and criteria for this Journal; among top 50%)

---

## Round 2 · Referee Report · Anonymous (Referee 3) · 2024-7-25

Report

This paper proposes the existence of quantum critical points in the phase diagram of lattice anti-ferromagnets described by $N_f = 4$ quantum electrodynamics in $2+1$ dimensions. An interesting aspect of this proposal is that the quantum critical points are between two identical phases of matter, a situation the authors refer to as that of an "unnecessary quantum critical points." I believe this is an interesting proposal, and that the paper should be published, but I find various parts of it a bit hard to follow in the current form. I have some concrete suggestions for how to improve it below.

Requested changes

  1. In I.B, first paragraph: it would be helpful to spell out what $G_{UV}$ is explicitly, in particular to define ${\cal T}$ and to list the generators of the square lattice space group. I assume these generators are listed in Table I in Appendix A, but a brief explanation of what these generators correspond to (lattice translations, rotations, etc.) would be very helpful. Maybe a drawing of the lattice would be useful.

  2. Paragraph containing eq. (5): it would be helpful to include a brief argument why the operator $\bar \psi_i \psi_j \Phi$ has dimension $\Delta_1 + 2 \sqrt{2}$ in the large $N_f$ limit. I assume it is because the lowest excitation of $\psi$ in the unit charge monopole background on the sphere has energy $\sqrt{2}$ (in units of the inverse radius of the sphere), but it would be good to see this explained more clearly.

  3. It would be helpful to connect the discussion at the end of Section II (paragraphs 4, 5, 6) about $q=0$ operators to tables II and III in Appendix A. In particular, which operators in these tables are fermion bilinears, which operators are quartic in the fermions, etc.?

  4. In the second paragraph of Section III, what does it mean for the two sets of 5 monopole operators to "couple" to four fermion terms or to fermion bilinear mass terms? Does the word "couple" mean that there's a term in the Lagrangian that contains products of monopole operators and two-fermion or four-fermion operaators? This doesn't make much sense, so the authors must mean something else by the word "couple". It would be great to clarify this point.

  5. Section III, end of 2nd paragraph: In the sentence "We label these operators as $n^a$ with $a = 1, \ldots , 5$", what do the words "these operators" refer to? Do they refer to the monopole operators or to the fermion bilinears?

  6. If the $n^a$ are fermion bilinears (as it is suggested in the 4th paragraph of Section III), which $SO(6)$ representation are the operators $n^a$ part of when $\lambda = 0$? Are they part of the ${\bf 15}$, which under the decomposition $SO(6) \to SO(5)$ becomes ${\bf 10} + {\bf 5}$, with the ${\bf 5}$ being the $n^a$? It would be great to clarify this point.

  7. Section IV, second paragraph: it would be nice to give more details supporting the idea that, in the absence of monopoles, the fermions of QED$_3$ are the $2 \pi$ vortices of an order parameter carrying charge-1 under $U(1)_\text{top}$. Where does this come from? Why are such vortices not gauge-invariant? Why do they transform in a projective representation of $SO(6)$ (namely the ${\bf 4} + \bar {\bf 4}$)? I think more explanation is needed.

  8. When writing $\bar \psi M \psi$ in the 2nd paragraph of Section IV, do the authors mean that $M$ is an $\mathfrak{su}(4)$ matrix instead of an $SU(4)$ matrix (i.e.~Lie algebra vs.~group element)? This should be made clear.

  9. In the second paragraph of Section IV, regarding the second to last sentence, if I'm understanding it correctly: why is it that if the leading operator that breaks a symmetry has a lower scaling dimension than the leading operator that doesn't break the symmetry, then one should expect the symmetry to be broken? It would be great to have an explanation, or to clarify the statement if I misunderstood it.

  10. Is there a difference between ${\bf n}$ in eq. (8) and $\hat n$ in Section IV.A? If there is, the authors should explain the difference; if not, it would be better I think to use the same notation everywhere.

  11. After eq. (10) it is mentioned that the operator in (10) is part of the symmetric traceless representation of SO(6). Which symmetric traceless representation? The rank-two one, namely the ${\bf 20}'$? It would be great to make this explicit.

  12. Shouldn't eq. (10) have $2(n_1^2 + n_2^2 + n_3^2) - 3 (n_4^2 + n_5^2)$ instead of $n_1^2 + n_2^2 + n_3^2 - n_4^2 - n_5^2$ so that it is a traceless polynomial?

  13. Third paragraph of Section IV.A: what is the evidence that the RG flow diagram is that in Figure 2? This fact is just stated with no evidence provided as far as I can tell, so I think the authors should improve the explanation here.

  14. Throughout the paper, the authors should change the notation for the rank-two symmetric traceless tensor representation of $SO(6)$ from ${\bf 20}$ to the more common notation ${\bf 20}'$. (The irrep the authors refer to is usually called the ${\bf 20}'$; the ${\bf 20}$ is usually a different irrep of $SU(4)$.)

  15. Appendix A: It would be good to explain in more detail where the assumptions for the triangular and Kagome lattices come from. As an alternative, one can remove the mention of these lattices from the Appendix since it does not seem immediately relevant to the present paper.

  16. It would be good to explain in more detail how Table I was derived. It seems very mysterious in its current form.

  17. Appendix A, 2nd paragraph, regarding ''$({\bf 20}, 0)$ and $({\bf 84}, 0)$ are singlets in any lattice QED$_3$ simulation.'' Do the authors mean "are singlets" or "contain singlets"?

  18. The allowed operators in the third columns of Tables II, III, IV do not obey the right symmetry and tracelessness conditions. For example, in Table II, the operator $O_1^\dagger O_1 + O_3^\dagger O_3$ is not traceless; instead, the traceless operator should be $O_1^\dagger O_1 + O_3^\dagger O_3 - \frac 12 (O_2^\dagger O_2 + O_4^\dagger O_4 + O_5^\dagger O_5 + O_6^\dagger O_6)$.

  19. If I understand correctly, I think the "allowed" operators in the third columns of Tables II, III, IV are not all the allowed operators. For instance, in the bottom right box of Table II, it seems to me that we can also have $O_2^\dagger O_2 - \text{trace}$, $O_4^\dagger O_4 - \text{trace}$, etc. It would be great if the authors could comment on this point (or correct me if I'm wrong).

  20. At the end of the Appendix, various estimates for scaling dimensions are given. The authors should give a reference or explain how they are derived.

  21. A general comment: the authors propose that $N_f = 4$ QED$_3$ can be found in the phase diagram of square lattice anti-ferromagnets. But as far as I can tell, the arguments in Sections III and IV involve continuum field theory, in particular QED$_3$ and its deformations. Do I understand correctly that the continuum field theory picture is a good approximation only if the parameters $\lambda$ and $\kappa$ are close to the origin in Figure 2? If so, is it obvious that this range of parameters can be accessed from the lattice Hamiltonian? If not, where is the continuum approximation valid?

Recommendation

Ask for minor revision

---

## Round 3 · Referee Report · Anonymous (Referee 2) · 2024-11-2

Report

The authors have addressed my questions, I am happy to recommend it for publication.

Recommendation

Publish (easily meets expectations and criteria for this Journal; among top 50%)

---

## Round 3 · Referee Report · Anonymous (Referee 1) · 2024-12-17

Report

I thank the authors for responding to my remarks (comments made by Referee # 1 in their reply). I am happy with most replies. Unfortunately I am still not satisfied about the discussion in response to one of my questions which led to Change 12.

First, I am not following the usage of the word slower. If operator $O_1$ has a higher scaling dimension than $O_2$, in which sense is the two-point correlation function $1/x^{2\Delta_{O_1}}$ "slower" than $1/x^{2\Delta_{O_2}}$? If anything it's faster, since it's decreasing faster. I think the use of "slower" is quite confusing. Can you replace the use of the word "slower" by just saying which operator has a larger scaling dimension?

Second, if these two operators did have unequal scaling dimensions, I don't see why this should imply anything about breaking symmetry, even at a heuristic level. Has this argument ever been invoked already in the literature? If yes, a reference would be welcome. If no, a more detailed explanation of heuristic would be very interesting to have, perhaps in a footnote.

Third, Ref. [10] states that the flavor and singlet mass operators actually have the same scaling dimension in the large $N_f $limit, to all orders in $1/N_f$ expansion ([10],App.D). This seems to contradict to what is written in the current paper in the second paragraph of Section IV, which seems to imply that the dimensions are actually unequal. (Also since Ref .[10] does not compute the dimensions but cites another paper for the calculation, that paper also needs to be cited.)

Recommendation

Ask for minor revision

---

## Round 3 · Referee Report · Anonymous (Referee 3) · 2024-12-24

Report

I thank the authors for addressing my comments! I noticed a few small things in the revision:

  1. Very minor: there is an instance of "there is are" in Section III.

  2. A couple of lines afterwards, the authors say "we will focus on the ordering of the particular set of 5 operators". I find the meaning of the word "ordering" a bit unclear; it would be great to rephrase. For instance, does it mean "condensation" / "acquiring an expecation value"?

  3. I find some aspects of the description of the various SO(6) representations in the Appendix still confusing / incorrect. In particular:

  4. in Table III, the expressions for the operators transforming in the 105, 175 are definitely incorrect. The operator that is supposed to transform in 105 is not symmetric in $(i, j, k, l)$, which is a requirement of this representation. The operator that is supposed to transform in 175 does not vanish when contracted with $\delta^{ij}$ which I think should be the case.

  5. in Table IV, the expression for the operator transforming in the 50 is incorrect because it is not symmetric in $(i, j, k)$.

  6. in the text it is mentioned that the operator transforming in the 50 is the traceless part $O_{ijk}^\dagger - \delta_{ij} \sum_j O_{jjk}^\dagger / 6$. This formula is incorrect because it is not symmetric in $(i, j, k)$.

  7. The phrase "minus traces and symmetric part" used in Tables III and IV is unclear. It is not clear whether one should symmetrize or remove the symmetric part. It would be better to give an actual formula like for the other cases.

I would be very happy to recommend the paper for publication after these issues are addressed.

Recommendation

Ask for minor revision

---

## Round 3 · Author Response

Dear Editor, Thank you for the referee report on our manuscript entitled “Dirac spin liquid as an “unnecessary” quantum critical point in square lattice antiferromagnets”, which we hereby resubmit to SciPost Physics Core. We appreciate the positive recommendation for publication as well as the questions raised by the referee. We will address these questions in a point-by-point response below. Having addressed all issues raised, we believe that the revised manuscript is now suitable for publication in SciPost Physics Core. Yours sincerely, Yunchao Zhang, Xue-Yang Song, and T. Senthil

Point-by-point response to the comments made by Referee # 1: Referee: This paper deals argues that QED3 with Nf = 4 fermions can be realized as an ”unnecessary” critical point sitting within the Neel phase of 2 + 1 D quantum antiferromagnets on the square lattice (and also within the VBS phase). ”Unnecessary” here means that on both sides of the codimension one critical manifold we find the same phase. Their argument starts by identifying the relevant operator which has to be tuned at criticality (charge-1 monopole). Then they argue that for any size of the perturbing monopole coupling the RG flow terminates in the same theory, SO(5) sigma model with a WZW term, Eq. (8). Then they discuss how this picture changes in presence of (dangerously) irrelevant perturbations arising from the microscopic symmetry. This is an interesting paper and I think it should be published. I have however a few questions and requests. Response: We are grateful for the positive evaluation of our work and for recommending publication in SciPost Physics Core. We next address the questions and requests of the referee. Referee: -Consider the RG flows (6) with the irrelevant terms ... set to zero. Is the RG flow with λ positive and negative supposed to be identical for all distances or only the IR fixed point (8) is supposed to be identical? The first possibility would be realized e.g. when perturbing the Ising model by the magnetic field perturbation, as is trivial to show since the sign of the coupling is flipped by a Z2. The second possibility is much more nontrivial. Whichever it is, it would be worth pointing out explicitly. Response: The RG flow for positive and negative λ are the same as the sign of λ is flipped by an SO(6) rotation. We have clarified this in Change 16. Referee: - p.3 last but one paragraph “hence must be irrelevant”. “Must be” sounds confusing here. Can the authors rephrase it more explicitly, e.g. that they believe it to be irrelevant based on this evidence and you will assume it so? (This is in line with the description of ∆2 above) Response: We have rephrased this in Change 4. Referee: - p.4 second paragraph. “One set of 5 will couple to four fermion terms” “orthogonal set of 5 operators that couple to the adjoint” I don’t understand what the authors mean by these phrases. Could be some jargon that I’m not aware of. I’d be grateful for more details here. Response: To be more clear, after λ̸ = 0, there will be symmetry allowed couplings between the 10 linear combinations of monopole and antimonopoles with four fermion and fermion billinear terms. This is clarified in Change 7. Referee: - p.5 first paragraph “has slower correlations” Can the authors explain what “slower” means here and why this implies the expectation that flavor symmetry is broken? Response: The slower correlations refers to the slower power law correlations of the flavor mass operator due to its higher scaling dimension. The argument that this implies broken flavor symmetry is heuristic and may not be true away from the critical point. We clarify this point in Change 12. Referee: - second column “supported by searches using the conformal bootstrap”. Here and elsewhere the authors demonstrate familiarity with the conformal bootstrap results, which is great. Can they provide some references here, and they do in other places of the paper? Response: We have added relevant references in Change 14. Referee: - p.6 sexond→second Response: We have corrected this typo in Change 20. Referee: - Appendix A. Are results in tables I,II,III,IV,V new? A citation or , alternatively, at least some details on the derivation would be needed. Response: Table I does not contain new results and was derived in a previous paper. We have added the appropriate citation in Change 21. The results in tables II-V are new. Using that the single monopole is an SO(6) vector, higher monopole/antimonopole operators correspond exactly to higher-indexed SO(6) tensors. This is clarified in Change 24. Referee: - Ref. [52] - “to a tricritical point” Several recent papers providing evidence for tricritical point include 2405.06607, 2405.04470, 2307.05307. I believe a reference these and other relevant works here would be appropriate, to properly balance the cited evidence for the complexed fixed points. Response: We thank the referee for pointing out these recent works. We have added references to all of these in Change 15. 4 Point-by-point response to the comments made by Referee # 2: Referee: The manuscript proposed Dirac spin liquid (DSL) as an unnecessary quantum critical point in square lattice antiferromagnets. The main idea is based on the conjecture that, deforming DSL with the relevant 2π monopole will trigger a RG flow towards the SO(5) DQCP, which will further flow to an ordered phase due to some dangerously irrelevant operator. Therefore, DSL can in principle appear as a critical point, inside the same ordered phase, i.e. Neel order or valence bond solid. The proposal is interesting, and would be helpful for future experimental or numerical search of DSL inside the more conventional ordered phase. I am happy to recommend this paper. Response: We thank the referee for the careful reading of our manuscript and for recommend- ing publication of it. Referee: 1. In the second paragraph of Sec. IV, the authors wrote “Then the fermions of QED3 can be viewed as the basic 2π vortices...”. I am not sure I understand this statement, maybe it is good to elaborate it a bit. Response: This is because the charge-1 objects of U (1)top,e.g. an elementary monopole, will acquire 2π Berry phase when moving around a fermion. We clarified in change 11. Referee: 2. Page 6, first paragraph, there is a typo, “The QED3 CFT will then describe a sexond...” should be second. Response: We have corrected this typo in Change 20.

Point-by-point response to the comments made by Referee # 3: Referee: This paper proposes the existence of quantum critical points in the phase diagram of lattice anti-ferromagnets described by NF = 4 quantum electrodynamics in 2 + 1 dimensions. An interesting aspect of this proposal is that the quantum critical points are between two identical phases of matter, a situation the authors refer to as that of an “unnecessary quantum critical points.” I believe this is an interesting proposal, and that the paper should be published, but I find various parts of it a bit hard to follow in the current form. I have some concrete suggestions for how to improve it below. Response: We appreciate the positive evaluation of our work and for all the improvements suggested. We next address all the requested modifications. Referee: 1. In I.B, first paragraph: it would be helpful to spell out what GU V is explicitly, in particular to define T and to list the generators of the square lattice space group. I assume these generators are listed in Table I in Appendix A, but a brief explanation of what these generators correspond to (lattice translations, rotations, etc.) would be very helpful. Maybe a drawing of the lattice would be useful. Response: We have made a revision in Change 2, spelling out exactly the square lattice space group and the action of time reversal on the fermions and monopoles. Referee: 2. Paragraph containing eq. (5): it would be helpful to include a brief argument why the operator ψiψj Φ has dimension ∆1 + 2√2 in the large Nf limit. I assume it is because the lowest excitation of ψ in the unit charge monopole background on the sphere has energy √2 (in units of the inverse radius of the sphere), but it would be good to see this explained more clearly. Response: Indeed, the scaling dimension being ∆1 + 2√2 in the large Nf limit is derived from exciting a mode from the n = −1 to n = 1 Landau level and then applying the state-operator correspon- dence. We have made Change 3 to include this explanation in the revised manuscript. Referee: 3. It would be helpful to connect the discussion at the end of Section II (paragraphs 4, 5, 6) about q = 0 operators to tables II and III in Appendix A. In particular, which operators in these tables are fermion bilinears, which operators are quartic in the fermions, etc.? Response: The fermion bilinears correspond to a monopole-antimonopole pair. Four fermion terms more complicated, as some of them are captured by four monopole-antimonopole terms while others require higher order monopole-antimonopole terms. We made Change 5 in the main text and Change 25 in the appendix to clarify this point and further emphasize the connection between the billinear/quartic fermion terms and composite monopole operators. Referee: 4. In the second paragraph of Section III, what does it mean for the two sets of 5 monopole operators to ”couple” to four fermion terms or to fermion bilinear mass terms? Does the word “couple” mean that there’s a term in the Lagrangian that contains products of monopole operators and two-fermion or four-fermion operaators? This doesn’t make much sense, so the authors must mean something else by the word “couple”. It would be great to clarify this point. Response: By couple, we mean that once the trivial monopole condenses (λ̸ = 0), there will be symmetry allowed terms in the action that couple the 10 monopole operators to fermion billinear masses or four fermion terms. We clarify this point in Change 7. Referee: 5. Section III, end of 2nd paragraph: In the sentence “We label these operators as na with a = 1, · · · , 5”, what do the words “these operators” refer to? Do they refer to the monopole operators or to the fermion bilinears? Response: The operators na refer to the monopole operators and we clarify this in Changes 8 and 9. Referee: If the na are fermion bilinears (as it is suggested in the 4th paragraph of Section III), which SO(6) representation are the operators na part of when λ = 0? Are they part of the 15, which under the decomposition SO(6) → SO(5) becomes 10 + 5, with the 5 being the na? It would be great to clarify this point. Response: Note the na are the monopoles, which can be identified with the fermion billinears when λ̸ = 0. And the na operators are part of the 15 of SO(6), which descends to the 5 of SO(5). We note this point in Change 10. Referee: 7. Section IV, second paragraph: it would be nice to give more details supporting the idea that, in the absence of monopoles, the fermions of QED3 are the 2π vortices of an order parameter carrying charge-1 under U (1)top. Where does this come from? Why are such vortices not gauge-invariant? Why do they transform in a projective representation of SO(6) (namely the 4 + 4)? I think more explanation is needed. Response: That the fermions are taken to be vortices of the U (1)top comes from the fact that the charged objects of U (1)top, i.e. the monopoles, acquire a Berry phase when circling a fermion. The vortices, which are not local operators, do not have to be gauge invariant. The Dirac fermions transform as fundamental of SU (4) flavor symmetry, which is isomorphic to the double cover of SO(6). This is allowed as the Dirac fermions (the vortices of U (1)top) are not gauge invariant and can be understood as arising from the mixed ’t Hooft anomaly between SO(6) and U (1)top. Referee: 8. When writing ψM ψ in the 2nd paragraph of Section IV, do the authors mean that M is an su(4) matrix instead of an SU (4) matrix (i.e. Lie algebra vs. group element)? This should be made clear. Response: M should be valued in the Lie algebra, or in other words, an adjoint SU (4) matrix. We clarify this point in Change 12. Referee: 9. In the second paragraph of Section IV, regarding the second to last sentence, if I’m understanding it correctly: why is it that if the leading operator that breaks a symmetry has a lower scaling dimension than the leading operator that doesn’t break the symmetry, then one should expect the symmetry to be broken? It would be great to have an explanation, or to clarify the statement if I misunderstood it. Response: Compared to ψψ, the flavor mass operator has slower power law correlations arising from its higher scaling dimension. The argument that this implies broken flavor symmetry is heuristic and may not be true away from the critical point. We clarify this point in Change 12. Referee: 10. Is there a difference between n in eq. (8) and ˆn in Section IV.A? If there is, the authors should explain the difference; if not, it would be better I think to use the same notation everywhere. Response: There is no difference, and we have made Change 13 to make the notation consistent throughout. Referee: 11. After eq. (10) it is mentioned that the operator in (10) is part of the symmetric traceless representation of SO(6). Which symmetric traceless representation? The rank-two one, namely the 20′? It would be great to make this explicit. Response: The relevant representation is exactly 20′, and we have made this explicit in Change 17. Referee: 12.(rephrased) On eq. (10) being a traceless polynomial? Response: To be more precise, it should be traceless, which we revise in Change 18. Referee: 13. Third paragraph of Section IV.A: what is the evidence that the RG flow diagram is that in Figure 2? This fact is just stated with no evidence provided as far as I can tell, so I think the authors should improve the explanation here. Response: The RG flows result from our previous arguments that κ is dangerously irrelevant (and therefore perturbatively irrelevant near the fixed point) and λ is relevant. We clarify this in Change 19. Referee: 14. Throughout the paper, the authors should change the notation for the rank-two symmetric traceless tensor representation of SO(6) from 20 to the more common notation 20′. (The irrep the authors refer to is usually called the 20′; the 20 is usually a different irrep of SU (4).) Response: We have made this change throughout the revised manuscript in Change 1. Referee: 15. Appendix A: It would be good to explain in more detail where the assumptions for the triangular and Kagome lattices come from. As an alternative, one can remove the mention of these lattices from the Appendix since it does not seem immediately relevant to the present paper. Response: The assumptions come from work done in Ref. [41] and we have added this and other references in Change 22. We included these in the Appendix in order to compare with the assumptions that we must make for the DSL to be a non-fine tuned critical point on the square lattice. Referee: 16. It would be good to explain in more detail how Table I was derived. It seems very mysterious in its current form. Response: Table I was derived in a previous paper, Ref. [17], and we have added a citation in Change 21. Referee: 17. Appendix A, 2nd paragraph, regarding “(20, 0) and (84, 0) are singlets in any lattice QED3 simulation.” Do the authors mean “are singlets” or “contain singlets”? Response: To be more precise, we mean these representations contain singlets. We clarify this is in Change 21. Referee: 18. The allowed operators in the third columns of Tables II, III, IV do not obey the right symmetry and tracelessness conditions. For example... (omitted) Response: The operators in the third columns of tables II-V. should have all the traces removed (and obey the right symmetry conditions) as spelled out in their corresponding second columns. This is made explicit in Change 26. Referee: 19. If I understand correctly, I think the “allowed” operators in the third columns of Tables II, III, IV are not all the allowed operators. For instance, in the bottom right box of Table II, it seems to me that we can also have O† 2O2 − trace, O† 4O4 − trace, etc. It would be great if the authors could comment on this point (or correct me if I’m wrong). Response: This point is correct. The “allowed” operators in the third columns of tables II-V are only examples of allowed operators and not an exhaustive list. We clarify this in Change 26 by adding “such as...”. Referee: 20. At the end of the Appendix, various estimates for scaling dimensions are given. The authors should give a reference or explain how they are derived. Response: We have added relevant citations in Change 27. Referee: 21. A general comment: the authors propose that Nf = 4 QED3 can be found in the phase diagram of square lattice anti-ferromagnets. But as far as I can tell, the arguments in Sections III and IV involve continuum field theory, in particular QED3 and its deformations. Do I understand correctly that the continuum field theory picture is a good approximation only if the parameters λ and κ are close to the origin in Figure 2? If so, is it obvious that this range of parameters can be accessed from the lattice Hamiltonian? If not, where is the continuum approximation valid? Response: Indeed the controlled limit is valid for small λ, κ. In principle certain lattice Hamil- tonian will be described by this limit in the low energy. The details of the microscopics are inac- cessible from the field theory. Currently the existing evidence on prevalent models, e.g. J1 − J2 Heisenberg model, does not show consistent behavior of a DSL. It is definitely interesting to search for such phenomenology in other frustrated magnetic systems.

---

## Round 3 · List of Changes

List of changes made 1. Entire Manuscript All instances of the 20 representation of SO(6) are replaced by 20′. 2. Section I.B., Paragraph 1 For the lattice spin model, GU V = SO(3) ⋊ T × p4m, where p4m is the square lattice symmetry group, generated by the unit lattice translations T1,2, C4 rotation, and mirror re- flection Rx. Time reversal T acts as an antiunitary symmetry that reverses monopole flux and fermion spin. 3. Section II., Paragraph 3, footnote following ”In the large-Nf limit, the scaling dimension of this operator is ∆1 + 2√2...” The scaling dimension is derived from the state-operator correspondence. Such an oper- ator corresponds to an excited 2π lorentz singlet monopole. The leading-order operator of this kind results from exciting a single sphere Landau level in a unit charge monopole background from level n = −1 to n = 1, which has excitation energy of 2√2. 4. Section II., Paragraph 5 ...and hence will be irrelevant since the CFT is found in a simulation[41]. Based on this evidence, we assume this operator is irrelevant. 5. Section II., Last Paragraph Lastly, we recall, as mentioned earlier, that all fermion bilinears and higher-order fermion terms can be written in terms of monopole operators. In particular, the fermion billinears correspond to monopole-antimonople pairs while four fermion operators correspond to com- posites of higher order monopole-antimonopole pairs (Appendix A). 6. Section III., Paragraph 2 Further it reduces the flavor symmetry from SO(6)... 7. Section III., Paragraph 2 There is are symmetry allowed terms in the Lagrangian that couple one set of 5 to four- fermion terms and the remaining set of 5 to the adjoint fermion billinear masses after λ̸ = 0. However, we will focus on the ordering of the particular set of 5 operators that couple to the adjoint fermion billinear mass[44]. Footnote 44: The specific form of the coupling can be found in [16] 8. Section III., Paragraph 2, Last Sentence We label these monopole operators as na with a = 1, ..., 5. 9. Section III., Paragraph 4 The 5-component na operator, representing the non-singlet monopoles... 10. Section III., Paragraph 4, Last Sentence To be more precise, at λ = 0, the 15 fermion billinear masses transform as the 15 represen- tation of SO(6) which branches into 5 ⊕ 10 under SO(6) → SO(5). It is the 5 that can be identified with na once λ̸ = 0. 11. Section IV., Paragraph 2 This arises from the reciprocal 2π Berry phase... 12. Section IV., Paragraph 2 It is known from calculations of scaling dimensions[10] (from the large-Nf expansion) that the flavor mass operator ¯ψM ψ where M is an adjoint SU (4) matrix exhibits slower power law correlations than the singlet mass operator ¯ψψ. Due to these enhanced correlations at the QED3 fixed point, we expect flavor symmetry to be broken, although a more thorough analysis would require a consideration of the parameter space surrounding the fixed point. 13. Section IV., Paragraph 3 n replaced by ˆn. 14. Section IV., Last Paragraph, References following ...supported by searches using the con- formal bootstrap Added references 54-55. 15. Section IV., Last Paragraph, Footnote 56 Added references 65-67. 16. Section IV.A., Paragraph 1 Moreover, the sign of λ can be flipped by an SO(6) rotation (i.e. the center of the group) in the DSL. 17. Section IV.A., Paragraph 2 ...in the traceless symmetric tensor representation, 20′, of SO(6) and... 18. Section IV.A., Paragraph 2, Eq. 10 19. Section IV.A., Paragraph 3 As κ is dangerously irrelevant and λ is relevant, we can obtain the... 20. Section IV.A., Paragraph 3 Corrected typo “sexond” to “second”. 21. Appendix A., Paragraph 2 The transformation of the single charge monopoles under the UV symmetries was derived in Ref. [17] and is outlined in Table I. 22. Appendix A., Paragraph 4 Using results from Refs. [16, 17], Ref. [41] derived the above UV singlet operators that must be RG irrelevant in order for the DSL to be stable. We note the operators (1, 0) must be irrelevant in order for the QED3 to flow to a true CFT in the IR without fine tuning. Furthermore, (20′, 0) and (84, 0) contain... 23. Appendix A., Paragraph 5 Now let us make the same assumptions for the square lattice as Ref. [41] does for the stability of the Kagome lattice DSL and observe if any additional conditions are required in order for the square lattice DSL to not be a multicritical point. 24. Appendix A., Paragraph 5 As O• transforms as the single index vector representation of SO(6), we can analyze the composites of O to derive explicit tensor forms of each representation and examples of symmetry allowed operators. 25. Appendix A.1., Paragraph 1 We note that fermion bilinears correspond to the operators transforming as (15, 0) in Ta- ble II. The four-fermion term will have overlap with numerous sectors, including (15, 0), (20′, 0), and (84, 0) in Table III., in addition to the SO(6) representations 45 and 45 that are hosted by higher order monopole-antimonopole operators. 26. Appendix A., Tables II.-IV. Modified third column entries to include the subtraction of trace components and added “such as” to clarify that the given allowed operators are only some of the examples. 27. Appendix A., Final bullet points Added references to where the scaling dimensions are derived.

---

## Round 4 · Author Response

Dear Editor, Thank you for the updated referee reports and editorial recommendation on our manuscript entitled “Dirac spin liquid as an “unnecessary” quantum critical point in square lattice antiferromagnets”, which we hereby resubmit to SciPost Physics Core. Having addressed all issues raised, we believe that the revised manuscript is now suitable for publication in SciPost Physics Core. Yours sincerely, Yunchao Zhang, Xue-Yang Song, and T. Senthil

Point-by-point response to the comments made by Referee # 1: Referee: I thank the authors for responding to my remarks (comments made by Referee # 1 in their reply). I am happy with most replies. Unfortunately I am still not satisfied about the discussion in response to one of my questions which led to Change 12. First, I am not following the usage of the word slower. If operator O1 has a higher scaling dimension than O2, in which sense is the two-point correlation function “slower” ? If anything it’s faster, since it’s decreasing faster. I think the use of “slower” is quite confusing. Can you replace the use of the word “slower” by just saying which operator has a larger scaling dimension? Response: We thank the referee for the careful reading of our manuscript. Instead of using the word “slower”, we have clarified that the flavor mass operator has a lower scaling dimension than the singlet mass operator in Change 3. Referee: Second, if these two operators did have unequal scaling dimensions, I don’t see why this should imply anything about breaking symmetry, even at a heuristic level. Has this argument ever been invoked already in the literature? If yes, a reference would be welcome. If no, a more detailed explanation of heuristic would be very interesting to have, perhaps in a footnote. Response: The two operators do have unequal scaling dimensions as the flavor mass operator has a lower scaling dimension than the singlet mass operator. Therefore, at the QED3 fixed point, the dominant long-wavelength correlations will come from the flavor mass operator and, one can imagine that slightly away from the fixed point, a flavor mass may be likely to condense, thus breaking the flavor symmetry. The intuition is that the operator with the slower correlations at the CFT fixed point is more “almost ordered” and if there is a relevant perturbation, then it is more likely to freeze in the fluctuations of this operator. A more concrete example of this heuristic argument is to consider an array of spin-1/2 chains coupled together by antiferromagnetic interactions between nearest neighbor chains. Each spin chain has power law Neel and VBS correlations; the Neel correlations are enhanced over the VBS by a log factor, so the Neel is (slightly) more slowly fluctuating than the VBS. Now the inter- chain coupling is known to be relevant. The belief is that the relevant flow takes you to the Neel ordered state, rather than the VBS ordered state (at least so long as the interchain interaction is not frustrating). This line of reasoning should only be taken as a rough heuristic in the absence of methods to probe the parameter space around the fixed point. We have added these comments in Changes 3 and 4. Referee: Third, Ref. [10] states that the flavor and singlet mass operators actually have the same scaling dimension in the large Nf limit, to all orders in 1/Nf expansion ([10], App.D). This seems to contradict to what is written in the current paper in the second paragraph of Section IV, which seems to imply that the dimensions are actually unequal. (Also since Ref. [10] does not compute the dimensions but cites another paper for the calculation, that paper also needs to be cited.) Response: Ref. [10] has an erratum which shows the flavor and singlet mass operators have different scaling dimensions. We have added the correct reference to this erratum in Changes 3 and 13. We have also added a reference to Ref. [24], which does the original calculation for the large Nf scaling dimensions.

Point-by-point response to the comments made by Referee # 2: Referee: The authors have addressed my questions, I am happy to recommend it for publica- tion. Response: We are grateful for the positive evaluation of our work and for recommending publication in SciPost Physics Core.

Point-by-point response to the comments made by Referee # 3: Referee: I thank the authors for addressing my comments! I noticed a few small things in the revision: Very minor: there is an instance of “there is are” in Section III. Response: We thank the referee for a careful reading of the manuscript. This typo is corrected in Change 1. Referee: A couple of lines afterwards, the authors say “we will focus on the ordering of the particular set of 5 operators”. I find the meaning of the word “ordering” a bit unclear; it would be great to rephrase. For instance, does it mean “condensation”/“acquiring an expecation value”? Response: We use the word ordering to mean condensation or acquiring an expectation value. This is clarified in Change 2. Referee: I find some aspects of the description of the various SO(6) representations in the Appendix still confusing/ incorrect. In particular: -in Table III, the expressions for the operators transforming in the 105, 175 are definitely incorrect. The operator that is supposed to transform in 105 is not symmetric in (i,j,k,l), which is a requirement of this representation. The operator that is supposed to transform in 175 does not vanish when contracted with δij , which I think should be the case. Response: We thank the referee for an attentive reading of the appendix. The expressions for 175 and 105 are incorrect as the traces being subtracted were not properly symmetrized. We have corrected this error in Change 8. Referee: because it is not symmetric in (i,j,k,l). - in Table IV, the expression for the operator transforming in the 50 is incorrect - in the text it is mentioned that the operator transforming in the 50 is the traceless part. This formula is incorrect because it is not symmetric in (i,j,k,l) Response: The expression for 50 is incorrect as the trace part being subtracted was not properly symmetrized. This is corrected in Change 11. The corresponding expression for the 50 irrep in the text is also corrected in Change 6. Referee: The phrase “minus traces and symmetric part” used in Tables III and IV is unclear. It is not clear whether one should symmetrize or remove the symmetric part. It would be better to give an actual formula like for the other cases. Response: We have added the explicit formulas for these cases in Changes 10 and 12. Referee: I would be very happy to recommend the paper for publication after these issues are addressed. Response: We are appreciation the positive evaluation of our work and for recommending publication in SciPost Physics Core.

---

## Round 4 · List of Changes

List of changes made
1. Section III., Paragraph 2
There is are corrected to There are.
2. Section III., Paragraph 2
However, we will focus on the condensation of the particular set of 5 operators that couple
to the adjoint fermion billinear mass.
3. Section IV. Paragraph 2
A further clue on the nature of this condensate comes from examining the particle-hole
operator with the lowest scaling dimension at the QED3 fixed point: it is presumably these
operators that will acquire an expectation value once the monopole fugacity flows to strong
coupling. It is known from calculations of scaling dimensions[10,11,24](from the large-Nf
expansion) that the flavor mass operator¯
ψMψ, where M is an adjoint SU(4) matrix, has
a lower scaling dimension than the singlet mass operator¯
ψψ. Therefore, the dominant,
slowly decaying, long-wavelength correlations at the QED3 fixed point will arise from the
flavor mass operator¯
ψMψ. Due to these enhanced correlations at the QED3 fixed point, we
heuristically expect flavor symmetry to be broken.
4. Section IV. Paragraph 2, footnote
The intuition is is that the operator with the slower correlations at the CFT fixed point
is more “almost ordered” and if there is a relevant perturbation, then it is more likely to
freeze in the fluctuations of this operator, though a more thorough analysis would require a
consideration of the parameter space surrounding the fixed point. A more concrete example
of this heuristic argument is to consider an array of spin-1/2 chains coupled together by
antiferromagnetic interactions between nearest neighbor chains. Each spin chain has power
law Neel and VBS correlations; the Neel correlations are enhanced over the VBS by a log
factor, so the Neel is (slightly) more slowly fluctuating than the VBS. Now as the inter-chain
coupling is known to be relevant, the belief is that the relevant flow leads to the Neel ordered
state, rather than the VBS ordered state (at least so long as the interchain interaction is not
frustrating).
5. Appendix A.1., Paragraph 1
Note the (105,0) is the fully symmetric, traceless representation, while (84,0) contains all
the rest of the operators symmetric with respect to O→O†, minus the fully symmetric and
trace components.
6. Appendix A.2., Paragraph 2
Corrected the expression for the form of the 50 irrep.
7. Appendix, Table II
Corrected minor typo in the second column for the 15 irrep in Table II.
8. Appendix, Table III
Corrected the explicit tensor forms for the 105 and 175 irreps.
9. Appendix, Table III
Corrected minor typo in the second column for 20’ irrep in Table III.
10. Appendix, Table III
Provided explicit tensor form for the 84 irrep.
11. Appendix, Table IV
Corrected the explicit tensor form for the 50 irrep.
12. Appendix, Table IV
Provided explicit tensor form for the 64 irrep.
13. References
Added reference to Hermele, Senthil, and Fisher, Phys. Rev. B 76, 149906 (2007).

---

## Editorial Decision

published